# Efficient Machine Unlearning for Deep Generative Models by Mitigating Optimization Conflicts

## Abstract

Machine unlearning of deep generative model refers to the process of modifying or updating a pre-trained generative model to forget or remove certain patterns or information it has learned. Existing research on Bayesian-based unlearning from various deep generative models has highlighted low efficiency as a significant drawback due to two primary causes. Firstly, Bayesian methods often overlook correlations between data to forget and data to remember, leading to conflicts during gradient descent and much slower convergence. Additionally, they require aligning updated model parameters with the original ones to maintain the generation ability of the updated model, further reducing efficiency. To address these limitations, we propose an **E**fficient **B**ayesian-based **U**nlearning method for various deep generative models called **EBU**. By identifying the relevant weights pertaining to the data to forget and the data to remember, EBU only preserves the parameters related to data to remember, improving the efficiency. Additionally, EBU balances the gradient descent directions of shared parameters to adeptly manage the conflicts caused by the correlations between data to forget and data to remember, leading to a more efficient unlearning process. Extensive experiments on multiple generative models demonstrate the superiority of our proposed EBU.

## 1 Introduction

In recent years, there have been significant advancements in deep generative models, showcasing their ability to produce synthetic images of exceptional quality (Wei et al., 2022; Li et al., 2022; Nichol & Dhariwal, 2021). These models typically rely on large volumes of training data to effectively learn and generate high-quality outputs (Wang et al., 2022; Cai & Zhu, 2015). However, the use of unauthorized data for training can lead to issues such as data misuse and privacy breaches (Li et al., 2021; Deepa et al., 2022; Wang et al., 2023c), raising concerns about the potential for these models to generate misleading or inappropriate content (Heng & Soh, 2024; Fan et al., 2024). Consequently, there is an urgent need to develop methods for mitigating the influence of specific data on pre-trained generative models.

The machine unlearning concept (Bourtoule et al., 2021) is proposed to demonstrate the problem that requires the trained machine learning models *unlearn* from specific data instances. Significant efforts have been made to advance the field of machine unlearning (Gupta et al., 2021; Sekhari et al., 2021; Nguyen et al., 2022).When it comes to the unlearning of multiple deep generative models, prior research has tackled this task by proposing Bayesian-based unlearning methods, albeit with certain inefficiency limitations (Chen et al., 2021; Deepanjali et al., 2021; Schuhmann et al., 2022; Heng & Soh, 2024; Fan et al., 2024). We attribute the inefficiency of Bayesian-based machine unlearning methods to two primary causes. Firstly, they neglect correlations between data to remember and data to forget (Heng & Soh, 2024; Wang et al., 2023a; Fan et al., 2024), leading to conflicts during gradient descent and much slower convergence, thereby exacerbating inherent inefficiencies. Secondly, Bayesian-based unlearning methods require aligning updated model parameters with the original ones to maintain generative ability of updated model (Heng & Soh, 2024; Wang et al., 2023a), introducing additional inefficiencies, notably prolonging the unlearning process.

Given the inefficiency limitations of existing Bayesian-based unlearning methods (Chen et al., 2021; Deepanjali et al., 2021; Schuhmann et al., 2022; Fan et al., 2024), we seek to enhance the efficiency of Bayesian-based forgetting methods. We propose EBU to stress the two fold limitations of Bayesian-based unlearning methods. Firstly, we selectively retain memory of parameters crucial for data to remember (Sener & Koltun, 2018; Désidéri, 2012), while updating parameters associated with data to forget, preventing the alignment of the entire model parameters, thereby improving efficiency and accelerating the unlearning process. Moreover, considering the correlation between data to forget and data to remember, EBU balances gradient updates on shared parameters associated with both types of data. This balancing mitigates conflicting gradient descent directions, narrowing conflicts, and further enhancing the efficiency of the unlearning process. Notably, the proposed EBU substantially improves the efficiency of the unlearning process and lays the groundwork for more effective model adaptation in deep generative models.

We summary our main contributions in short as follows:

- We introduce EBU, a groundbreaking framework that dramatically enhances the efficiency of machine unlearning in deep generative models by effectively resolving the conflicts between forgetting and remembering processes. This innovation significantly improves both concept-wise and class-wise unlearning.

- EBU strategically preserves critical parameters tied to the data that must be remembered, making the unlearning process in Bayesian-based models far more efficient and streamlined than previous approaches.

- By incorporating a novel mechanism to balance gradient updates for forgetting and remembering, EBU accelerates the entire unlearning process, ensuring faster and more reliable performance.

- Extensive experiments across diverse datasets and generative models clearly demonstrate the superior performance of EBU, proving its effectiveness and efficiency compared to existing baseline methods.

## 2 RELATED WORK

**Machine unlearning for generative model** In recent years, the researchers have made efforts in unlearning of generative model. Several works proposed to unlearning from GANs by utilizing the discriminator (Kong & Chaudhuri, 2023; Chen et al., 2021; Sun et al., 2023), but these methods can't be applied to other generative models due to they only suit for paradigm of GANs. There are some works realizing unlearning of generative model by modifying the weights (Bau et al., 2020; Tarun et al., 2023), but it is still a challenge to accurately identify model weights associated to the forgetting tasks accurately. Some researchers proposed Bayesian based unlearning methods that can be applied to various generative models (Heng & Soh, 2024; Nguyen et al., 2020; Fan et al., 2024; Fu et al., 2022; 2021), but they need to trade off between forgetting the posterior distribution of data to forget and not entirely forgetting posterior distribution of the original training data to preserve the generative models' ability, causing conflicts due to the correlations between the data to forget and the original training data.

**Difference between EBU and prior methods** Our proposed EBU is a Bayesian based unlearning method and can be compatible with any generative models, but advanced with the existing Bayesian based unlearning methods (Heng & Soh, 2024; Nguyen et al., 2020; Fan et al., 2024), our method reduce the conflicts by balancing the gradient descent directions of the parameters shared by the forgetting and remembering processes. We identify these shared parameters by analyzing corresponding weight saliency maps during the unlearning process. In contrast to previous methods that use saliency maps to identify parameters for updates (Fan et al., 2024), our approach dynamically selects parameters specifically related to forgetting and remembering during the fine-tuning process. This dynamic selection allows us to guide the gradient descent directions for forgetting and remembering separately, while keeping unrelated parameters unchanged. By preventing optimization conflicts between the two tasks, our method not only enhances unlearning efficiency but also ensures more precise parameter updates, leading to superior performance.

## 3 PRELIMINARY

### 3.1 PROBLEM DEFINITION

We give a brief introduction of the forgetting process of the deep generative model. Given a pre-trained generative model $G_{\boldsymbol{\theta}}$ with parameter $\boldsymbol{\theta}$ trained on the dataset $D = \{(X^n, Y^n)\}_{n=1}^{N}$, where $N$ is the number of categories in the dataset. Without accessing the training data, we generate forgetting set $D_f$ and remembering set $D_r$ using the model $G_{\boldsymbol{\theta}}$. Here, $D_f = \{(X_f^n, Y_f^n)\}_{n=1}^{N_f}$ denotes the set of data to forget and $D_r = \{(X_r^n, Y_r^n)\}_{n=1}^{N_r}$ denotes the set of data to remember, with $N_r + N_f \leq N$. In this context, $X_f^n$ and $Y_f^n$ denote the data to forget and corresponding labels for category $n$. $X_r^n$ and $Y_r^n$ denote the data to remember and the corresponding labels for category $n$. Our goal is to forget the assigned set $D_f$ from the pre-trained generative model $G_{\boldsymbol{\theta}}$ while keeping the generation quality of the remaining samples in $D_r$ by fine-turning the pre-trained deep generative model. The fine-tuned model $G_{\boldsymbol{\theta}^*}$ with parameters $\boldsymbol{\theta}^*$ is expected to forget all samples in $D_f$ while retaining the ability to generate samples conforming to the distribution $p_{\boldsymbol{\theta}^*}(D_r) \sim G_{\boldsymbol{\theta}^*}(X_r|Y_r)$ that is expected to align with the distribution of the data to remember $p(D_r)$. For clarity and ease of representation, all model parameters used in this paper are denoted as the set of their individual elements. For instance, $\boldsymbol{\theta} = \{\theta_1, \theta_2, \ldots, \theta_i, \ldots, \theta_k\}$, where $k$ represents the total number of elements within $\boldsymbol{\theta}$ and $\theta_i$ denotes the $i^{th}$ element of the model parameters.

### 3.2 MOTIVATION

We are interested in forgetting specified samples from a pre-trained diffusion model. Prior work attempted to forget data from a model to ensure the privacy in machine learning models by deleting only specific shards (Bourtoule et al., 2021), thereby forgetting these assigned shards. Moreover, building on the ideas from (Heng & Soh, 2024; Nguyen et al., 2020), they implemented forgetting of model by forgetting the posterior belief of data $D_f$ while not forgetting the posterior belief given the full data $D$. As suggested by (Heng & Soh, 2024), using Elastic Weight Consolidation (EWC) (Kirkpatrick et al., 2017) keeps the posterior belief of the full data $p_{\boldsymbol{\theta}}(D)$, preventing catastrophic forgetting. However, this approach hinders the forgetting process by maintaining the posterior belief of full data $p_{\boldsymbol{\theta}}(D)$. A more reasonable solution is to remember the posterior belief of $D_r$ while forgetting the posterior belief of $D_f$. The weights for forgetting $p_{\boldsymbol{\theta}}(D_f)$ and remembering $p_{\boldsymbol{\theta}}(D_r)$ are denoted as $\boldsymbol{\theta}_f$ and $\boldsymbol{\theta}_r$, respectively. Fast forgetting of $D_f$ can be achieved by keeping $\boldsymbol{\theta}_r$ consistent with its original values and leaving $\boldsymbol{\theta}_f$ unchanged. Moreover, the overarching unlearning process necessitates a delicate trade-off between forgetting $p_{\boldsymbol{\theta}}(D_f)$ while retaining aspects of $p_{\boldsymbol{\theta}}(D)$. The parameters affected by this trade-off are those within $\boldsymbol{\theta}_f \cap \boldsymbol{\theta} = \boldsymbol{\theta}_f$. When compared to the general unlearning approach, focusing solely on addressing conflicts within $\boldsymbol{\theta}_r \cap \boldsymbol{\theta}_f \subseteq \boldsymbol{\theta}_f$ results in a smaller negative impact on the unlearning process, thereby potentially accelerating it.

## 4 METHOD

We introduce a novel unlearning method EBU aimed at expediting the forgetting process of pre-trained deep generative models while preserving the quality of the generated images. We present a comprehensive theoretical analysis to underpin our proposed EBU in Section 4.1, elucidating the underlying rationale behind our method's efficacy in resolving conflicts inherent in the unlearning process. In particular, EBU comprises two key components. First, we introduce a partial parameter alignment method in Section 4.2, which significantly enhances unlearning efficiency. Second, in Section 4.3, we propose an effective approach to mitigate optimization conflicts between remembering and forgetting, further accelerating the unlearning process.

### 4.1 THEORETICAL ANALYSIS

The optimization objective of the unlearning of deep generative models is to minimize the expected loss functions $\mathcal{L}_f$ and $\mathcal{L}_r$ over the distributions of forgetting dataset $p(D_f)$ and remembering dataset $p(D_r)$:

$$\mathcal{O}_1 = \min_{\{\boldsymbol{\theta}_f, \boldsymbol{\theta}_r\}} \mathbb{E}_{p(D_f)}[\mathcal{L}_f(X_f, \boldsymbol{\theta}_f)] + \mathbb{E}_{p(D_r)}[\mathcal{L}_r(X_r, \boldsymbol{\theta}_r)] \tag{1}$$

But the distributions $p(D_f)$ and $p(D_r)$ are unavailable, and one solution is to optimize the forgetting task and remembering task independently. However, it's important to note that the data to forget and the data to remember might be related. If handle the remembering and forgetting separately, we could lose important information because the parameters involved in both tasks might conflict with each other (Sener & Koltun, 2018). Thus training the data of forgetting and remembering simultaneously is another solution, which results in a new optimization task:

$$\mathcal{O}_2 = \min_{\boldsymbol{\theta}} \frac{1}{N_f + N_r} \sum_{i=1}^{N_f} \mathcal{L}_f(X_f^i, \boldsymbol{\theta}) + \sum_{i=1}^{N_r} \mathcal{L}_r(X_r^i, \boldsymbol{\theta}) \tag{2}$$

However, this training approach overlooks the distinction between the data intended for forgetting and the data intended for remembering. As a result, it may hinder the model from converging since the gradient of the first loss term and the second loss term in Equation (2) may update in opposite direction. In the context of deep generative model, according to the PAC-Bayesian theory (McAllester, 1998), there exists a error $\epsilon(N, \zeta) \geq 0$ with probability $1 - \zeta$ over independent draws monotonically decreasing with the training samples $N$ between the expected optimization objective and actual optimization objective (McAllester, 1998) (the detailed proof is provided in Appendix C). We can obtain the following proposition:

**Proposition 4.1.** *The bounds between the optimization objective $\mathcal{O}_1$ and $\mathcal{O}_2$:*

$$\begin{aligned} |\mathcal{O}_1 - \mathcal{O}_2| \leq &\mathbb{E}_{p(D_f)}[G_{\boldsymbol{\theta}_f}(X_f|Y_f) - G_{\boldsymbol{\theta}_e}(X_f|Y_f)] + \mathbb{E}_{p(D_r)}[G_{\boldsymbol{\theta}_r}(X_r|Y_r) - G_{\boldsymbol{\theta}_e}(X_r|Y_r)] \\ &+ \epsilon(N_f + N_r, \zeta) \end{aligned} \tag{3}$$

*where $\boldsymbol{\theta}_e$ denotes the optimal solution of Equation (2) and $G_{\boldsymbol{\theta}_{(\cdot)}}$ denotes the deep generative model with parameters $\boldsymbol{\theta}_{(\cdot)}$.*

Considering the differences and connections between the data to forget and the data to remember at the same time, we rewrite the optimization objective as:

$$\begin{aligned} \mathcal{O}^* = &\min_{\{\boldsymbol{\theta}_f, \boldsymbol{\theta}_r, \boldsymbol{\theta}_e\}} \frac{1}{N_f} \sum_{i=1}^{N_f} \mathcal{L}_f(X_f^i, \boldsymbol{\theta}_f) + \frac{1}{N_r} \sum_{i=1}^{N_r} \mathcal{L}_f(X_r^i, \boldsymbol{\theta}_r) \\ &\text{subject to } |\mathbb{E}_{p(D_f)}[G_{\boldsymbol{\theta}_f}(X_f|Y_f) - G_{\boldsymbol{\theta}_e}(X_f|Y_f)]| \leq \xi \\ &\qquad\qquad |\mathbb{E}_{p(D_r)}[G_{\boldsymbol{\theta}_r}(X_r|Y_r) - G_{\boldsymbol{\theta}_e}(X_r|Y_r)]| \leq \xi \end{aligned} \tag{4}$$

The constant $\xi$ controls the closeness between functions. A larger $\xi$ allows functions to be more task-specific. The expectations in Equation (4) can't be calculated directly due to the lack of the accessibility to the probability distribution $p(\cdot)$. But if the function is Lipschitz in the parameterization, the distance between the functions can be measured by the distance between parameters (Cervino et al., 2021), thus we have another proposition to estimate the expectations:

**Proposition 4.2.** *The two generative models can be seen as two parametric functions, thus it has:*

$$\begin{aligned} \mathbb{E}_{p(D_f)}[G_{\boldsymbol{\theta}_f}(X_f|Y_f) - G_{\boldsymbol{\theta}_e}(X_f|Y_f)] &\leq L|\boldsymbol{\theta}_f - \boldsymbol{\theta}_e|, \\ \mathbb{E}_{p(D_r)}[G_{\boldsymbol{\theta}_r}(X_r|Y_r) - G_{\boldsymbol{\theta}_e}(X_r|Y_r)] &\leq L|\boldsymbol{\theta}_r - \boldsymbol{\theta}_e| \end{aligned} \tag{5}$$

*where $L$ is a constant that decides the scope.*

Thus through imposing constraints on parameters, the Equation (4) can be solved. The forgetting and remembering tasks relate to part parameters $\boldsymbol{\theta}_f$ and $\boldsymbol{\theta}_r$, we present how to select and modify the corresponding $\boldsymbol{\theta}_f$ and $\boldsymbol{\theta}_r$ to unlearn from deep generative model in Section 4.2.

## 4.2 PARTIAL PARAMETER ALIGNED BAYESIAN UNLEARNING

Existing works (Chen et al., 2021; Deepanjali et al., 2021; Schuhmann et al., 2022) for unlearning of deep generative models need to *fully align* the parameters between the unlearned model and original pre-trained model. This leads to highly inefficient unlearning. In this section, we propose a technique which only needs to *partially align* the parameters between the unlearned model and original pre-trained model, thereby improving the efficiency significantly. Specifically, in Section 4.1, we define the optimization objective of unlearning from deep generative models in Equation (4), but the expectations can't be calculated directly. As described in Proposition 4.2, the expectations can be estimated by accessing the corresponding parameters, then this optimization objective can be solved. In this section, we propose efficient ways to select the corresponding parameters $\boldsymbol{\theta}_f$ and $\boldsymbol{\theta}_r$, thereby solving the Equation (4) to unlearn from the deep generative models efficiently.

**Learning objective for the data to forget** To forget $D_f$ from the pre-trained deep generative model $G_\theta$, we need to make the posterior distribution $p(X_f|\theta^*, Y_f)$ far from the real distribution $p(X_f|Y_f)$ as much as possible. One way is to make the unlearned model to generate samples from a *mandatory distribution* $\tilde{q}(X_f|Y_f)$, which should be different from the real distribution (Heng & Soh, 2024) and usually set it to a standard Gaussian distribution. We minimize the KL divergence between the generated sample distribution from the unlearned model and the designated mandatory distribution during the fine-turning process:

$$\mathcal{L}_f = \mathbb{E}_{p(Y_f)} D_{KL}(p(X_f|\theta^*, Y_f)||\tilde{q}(X_f|Y_f)) \tag{6}$$

we denote the class distributions in $D_f$ as $p(Y_f)$. In this way, the fine-tuned generative model $G_{\theta^*}$ will change the learned distribution of the forgotten set to *mandatory distribution* after fine-tuning, so as to achieve the purpose of making the model forget. Accordingly, the parameters $\theta_f$ related to forgetting task should be far from the original parameters.

**Learning objective for the data to remember** During the forgetting process, it's necessary to strengthen the memory of generative model for the data in the remember set $D_r$. We thus replay the data from $D_r$ during the forgetting process to keep the model's ability to generate the data to remember. By replaying data in $D_r$, the updated posterior distribution of $D_r$ denoted as $p(X_r|\theta^*, Y_r)$ is forced to approach to the original posterior distribution $p(X_r|\theta, Y_r)$, preserving the model's ability to generate the data to remember. To replay data from $D_r$, we define an optimization function as:

$$\mathcal{L}_r = \mathbb{E}_{p(Y_r)} D_{KL}(p(X_r|\theta^*, Y_r)||p(X_r|\theta, Y_r)) \tag{7}$$

through this optimization function, the generated sample distribution of $D_r$ with the unlearned model $G_{\theta^*}$ will be consistent with the distribution of the data to remember on the original model $G_\theta$.

During the optimization process, the posterior distribution of $D_r$ on model $G_{\theta^*}$ should be close to the posterior distribution on $G_\theta$, whereas on $D_f$ the situation is reverse. That is we should keep the model parameters $\theta_r$ related to remembering close to the corresponding parameters in $\theta$ and $\theta_f$ related to forgetting far from the corresponding parameters in $\theta$. We use $\theta_r^{-f}$ to represent the remaining parameters of $\theta_r$ after removing the overlap with the $\theta_f$, and $\hat{\theta}_r^{-f}$ denotes the elements of the original pre-trained model parameters $\theta$ with same parameter index of the elements in $\theta_r^{-f}$:

$$\hat{\theta}_r^{-f} = \theta \times \mathcal{O}(\theta_r^{-f}), \ \theta_r^{-f} = \theta_r \backslash (\theta_r \cap \theta_f) \tag{8}$$

where $\mathcal{O}(\cdot)$ sets the non-zero values in the set to 1 and leave the zero values unchanged, the symbol $\times$ denotes element-wise multiplication and $\backslash$ denotes the operation of obtaining complement set.

Motivated by the continue learning algorithm EWC (Kirkpatrick et al., 2017), we make $\theta_r$ close to $\theta$ at each gradient step to enforce the model remembering $D_r$, and the optimization function of remembering $D_r$ can be further rewritten as follows,

$$\mathcal{L}_r^c = \mathcal{L}_r + \frac{\gamma}{2} \sum_i F_i (\theta_{r,i}^{-f} - \hat{\theta}_{r,i}^{-f})^2 \tag{9}$$

where $\gamma$ is a constant that regularizes the degree that make the new parameters close to the old parameters, $i$ denotes the index of the element, and $F$ is the set of diagonal elements of the fisher information matrix calculated on $\theta$. Existing work (Chen et al., 2021; Heng & Soh, 2024) aligns the entire parameters $\theta$ and $\theta^*$. In contrast, our work only needs to partially align $\theta_{r,i}^{-f}$ and $\hat{\theta}_{r,i}^{-f}$ in Equation (9), thereby speeding up the unlearning process.

## 4.3 Optimization Conflicts Mitigation

Forgetting data $D_f$ from the generative model changes part of the parameters, while replaying data $D_r$ to the model also changes part of parameters, we denote the two parts of parameters as $\theta_f$ and $\theta_r$ respectively. However, data from different categories may exhibit similar patterns, potentially influencing a shared set of model parameters, *i.e.*, $\theta_f \cap \theta_r$. As shown in Figure 1, the changed parameters $\theta_f$ and $\theta_r$ partially overlap. When optimizing $\mathcal{L}_r$ and $\mathcal{L}_f$ simultaneously, the gradient update directions for forgetting and replaying conflict, hindering the model's ability to forget or remember effectively and slowing convergence. To address this, a trade-off strategy is needed to balance forgetting and remembering during optimization. In this section, we propose efficient methods to mitigate these conflicts and accelerate the unlearning process.

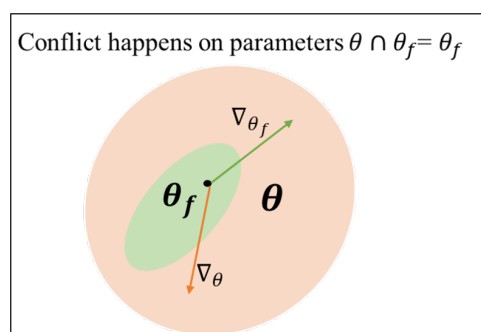 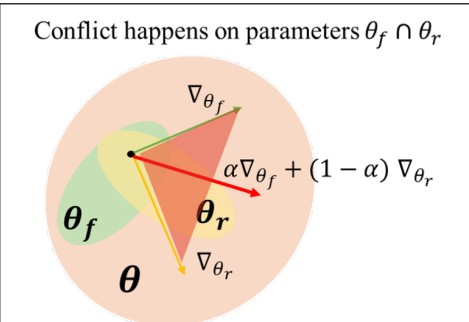

(a) Existing unlearning methods (Nguyen et al., 2020; Heng & Soh, 2024)

(b) Our proposed unlearning method EBU

Figure 1: In Figure 1(a), we illustrate the conflict inherent in existing unlearning methods, where the conflict spans across all $\boldsymbol{\theta}_f$ when forgetting specific data while not entirely forgetting the entire dataset. Figure 1(b) depicts our proposed unlearning method, which narrows the scope of conflicted parameters to $\boldsymbol{\theta}_f \cap \boldsymbol{\theta}_r$, we also employ a trade-off parameter $\alpha$ to balance the conflicted gradient descent directions, mitigating their negative impact on the unlearning process.

**Parameters selection**    During the unlearning process, the parameters that are closely related to the forgetting and remembering tasks will exhibit larger gradients compared to the irrelevant parameters. This observation allows us to identify the parameters pertinent to each task based on their gradients at each gradient step. We denote the entire set of parameters of the deep generative model as $\boldsymbol{\theta}_t^*$ at the $t$-th gradient step. The optimization functions for forgetting and remembering at this step are represented by $\mathcal{L}_f^t$ and $\mathcal{L}_r^t$, respectively. By analyzing these gradients, we can effectively discern which parameters need to be adjusted for efficient unlearning.

We utilize a constant $\sigma \in [0, 1]$ to determine the proportion of selected parameters based on the gradient values $\nabla_{\boldsymbol{\theta}_t^*} \mathcal{L}_f^t$. By applying this threshold, we can identify the parameters $\boldsymbol{\theta}_f^t$ associated with forgetting at the $t$-th step by selecting the top $\sigma$ proportion of gradient values.

$$\boldsymbol{\theta}_f^t = \boldsymbol{\theta}_t^* \times \mathcal{O}(\text{top-}\sigma(\nabla_{\boldsymbol{\theta}_t^*} \mathcal{L}_f^t)) \tag{10}$$

We fill zeros in the empty spaces to maintain the size of the selected parameters consistent with the original parameters. This ensures that the overall structure of the parameter set remains intact while allowing us to focus on the relevant parameters associated with the forgetting task. By preserving the dimensions of the parameter set, we can seamlessly integrate these updates into the model without disrupting its architecture.

**Gradient Modulation**    To effectively resolve conflicts arising from shared parameters, we propose a straightforward trade-off method that balances the effects of forgetting and remembering by averaging their gradients. This approach is particularly effective because it neutralizes the competing influences of both processes, preventing one from undermining the other.

By defining $\boldsymbol{\theta}_f^{t,r} = \boldsymbol{\theta}_f^t \cap \boldsymbol{\theta}_r^t$ to represent the parameters in $\boldsymbol{\theta}_f^t$ that overlap with those in $\boldsymbol{\theta}_r^t$, we ensure that we focus on the shared parameters that are crucial for both forgetting and remembering. Averaging the gradients for these overlapping parameters allows us to update them in a way that accommodates the requirements of both tasks simultaneously:

$$\nabla_{\boldsymbol{\theta}_f^{t,r}} \mathcal{L}_f^t := \alpha \nabla_{\boldsymbol{\theta}_f^{t,r}} \mathcal{L}_r^t \oplus (1 - \alpha) \nabla_{\boldsymbol{\theta}_r^{t,f}} \mathcal{L}_f^t \tag{11}$$

where $\oplus$ denotes the element-wise sum operation applied to elements with the same index, while $\alpha \in (0, 1)$ serves as a trade-off constant. Adjusting $\alpha$ allows us to steer the gradient descent directions of the parameters shared between $\boldsymbol{\theta}_f^t$ and $\boldsymbol{\theta}_r^t$ (we delve into the effect of different $\alpha$ values on the unlearning process in Section 5.5).

This method effectively mitigates optimization conflicts by ensuring that updates made for forgetting do not completely override the updates for remembering, and vice versa. As a result, we maintain a balanced influence on the shared parameters, which leads to a more coherent unlearning process.

This dual consideration enhances overall model efficiency and ensures that important information is preserved while unwanted knowledge is successfully removed.

During the optimization process, it is crucial to effectively erase the data we wish to forget from the deep generative model. To achieve this, we adjust the gradient descent to be more focused on forgetting, thereby accelerating the forgetting process. We denote $\mathcal{L}_r^{c,t}$ as the $t$-th instance of the forgetting loss $\mathcal{L}_r^c$ (refer to Equation Equation (9)).

$$\nabla_{\boldsymbol{\theta}_t^*}(\mathcal{L}_f^t + \mathcal{L}_r^{c,t}) := \nabla_{\boldsymbol{\theta}_t^*}(\mathcal{L}_f^t + \mathcal{L}_r^{c,t}) + \nabla_{\boldsymbol{\theta}_f^t}\mathcal{L}_f^t \tag{12}$$

This targeted approach ensures that the model prioritizes updates that facilitate the removal of unwanted information while maintaining the integrity of the data we intend to remember.

Consequently, the overall optimization process can be summarized as:

$$\boldsymbol{\theta}_{t+1}^* \leftarrow \boldsymbol{\theta}_t^* - \lambda\nabla_{\boldsymbol{\theta}_t^*}(\mathcal{L}_f^t + \mathcal{L}_r^{c,t}), \quad \boldsymbol{\theta}_0^* = \boldsymbol{\theta} \tag{13}$$

where $\lambda$ is the learning rate. This formulation highlights how the parameters are updated by taking into account both the forgetting and remembering objectives, ensuring a balanced approach that facilitates effective unlearning. The comprehensive algorithm of our proposed method is outlined in Algorithm 1.

---

**Algorithm 1** Unlearning Process of EBU

1: **Input:** Pre-trained model $G_{\boldsymbol{\theta}}$, data to retain $D_r = \{(X_r^n, Y_r^n)\}_{n=1}^{N_r}$, data to forget $D_f = \{(X_f^n, Y_f^n)\}_{n=1}^{N_f}$, desired distribution $\tilde{q}(X_f|Y_f)$, learning rate $\lambda$, initial parameter set $\boldsymbol{\theta}_0^* = \boldsymbol{\theta}$
2: **Output:** Fine-tuned model $G_{\boldsymbol{\theta}^*}$, $\boldsymbol{\theta}^* = \boldsymbol{\theta}_T^*$
3: **for** $t = 0$ to $T$-1 **do**
4:      Compute gradients $\nabla_{\boldsymbol{\theta}_t^*}\mathcal{L}_f^t = \partial\mathcal{L}_f^t/\partial\boldsymbol{\theta}_t^*$ and $\nabla_{\boldsymbol{\theta}_t^*}\mathcal{L}_r^t = \partial\mathcal{L}_r^t/\partial\boldsymbol{\theta}_t^*$
5:      Select parameter subsets $\boldsymbol{\theta}_f^t$ and $\boldsymbol{\theta}_r^t$ based on the top $\delta$ proportion values of $\nabla_{\boldsymbol{\theta}_t^*}\mathcal{L}_f^t$ and $\nabla_{\boldsymbol{\theta}_t^*}\mathcal{L}_r^t$ respectively.
6:      Identify overlapping parameters: $\boldsymbol{\theta}_f^{t,r} = \boldsymbol{\theta}_r^{t,f} = \boldsymbol{\theta}_f^t \cap \boldsymbol{\theta}_r^t$
7:      Calculate gradients for overlapping parameters: $\nabla_{\boldsymbol{\theta}_f^{t,r}}\mathcal{L}_f^t := \alpha\nabla_{\boldsymbol{\theta}_f^{t,r}}\mathcal{L}_f^t \oplus (1-\alpha)\nabla_{\boldsymbol{\theta}_r^{t,f}}\mathcal{L}_r^t$
8:      Update parameters: $\boldsymbol{\theta}_{t+1}^* \leftarrow \boldsymbol{\theta}_t^* - \lambda\nabla_{\boldsymbol{\theta}_t^*}(\mathcal{L}_f^t + \mathcal{L}_r^c), \ \nabla_{\boldsymbol{\theta}_t^*}(\mathcal{L}_f^t + \mathcal{L}_r^c) \leftarrow \nabla_{\boldsymbol{\theta}_t^*}(\mathcal{L}_f^t + \mathcal{L}_r^c) + \nabla_{\boldsymbol{\theta}_f^t}\mathcal{L}_f^t$
9: **end for**

---

## 5 EXPERIMENT

In this section, we demonstrate the ability of proposed EBU in assisting various deep generate models unlearning certain classes and concepts. We compare our method with the existing state-of-the-art unlearning baselines, highlighting the effectiveness of EBU.

### 5.1 EXPERIMENT SETTING

**Implements** We focus on two types of forgetting tasks: class-wise forgetting and concept-wise forgetting. We utilize two types of generative models: the pre-trained DDPM (Ho et al., 2020) and the Stable Diffusion (SD) model (Rombach et al., 2022) to assess the performance of our proposed method. The hyperparameters $\alpha$ and $\delta$ are set to 0.6 and 0.5, respectively. All experiments are conducted using 4 Nvidia V100 GPUs with 32 GB memory. More detailed experiment implements of both the class-wise unlearning and concept-wise unlearning tasks can be found in Appendix B.2.

**Baselines** We compare our proposed EBU with other five different state-of-the-art methods to evaluate the efficiency and fidelity of EBU. We choose three general unlearning methods: FT (Warnecke et al., 2021), GA (Thudi et al., 2022) and Retraining, two unlearning methods for deep generative methods: SA (Heng & Soh, 2024) and ESD (Gandikota et al., 2023). The detailed description and implements of the baseline methods are presented in Appendix B.

**Metrics** To evaluate the fidelity of unlearning methods for class-wise forgetting, we use two metrics: classification entropy (CE) for forgetting classes and remaining accuracy (RA) for the accuracy of the remaining classes in the unlearned model. For assessing efficiency, we consider unlearn time

(UT) and relearn time (RT). UT measures the time for the unlearning process, while RT counts the gradient updating steps needed for the unlearned model. We also use the Fréchet Inception Distance (FID) to assess the image quality of the classes to remember, aiming for minimal impact on their quality by the unlearned model. As for the concept-wise forgetting, we use two metrics: Clip Score (CS) (Hessel et al., 2021) and Nudity Score (NS) used in (Gandikota et al., 2023) to evaluate the forgetting performance of different unlearning methods.

## 5.2 CLASS-WISE FORGETTING

The class-wise forgetting is to unlearn the specified classes, we evaluate the class-wise forgetting performance of unlearning methods on pre-trained DDPM and SD.

**Main results on DDPM** We conduct experiments on CIFAR10, STL10 and CIIFAR100. The experiment results of DDPM on CIFAR-100 and STL10 are presented in Table 6, the experiment results on CIFAR10 are presented in Appendix D.4, multiple unlearning methods are applied to pre-trained DDPM to demonstrate the effectiveness of unlearning methods. Here we present the experiment results of unlearning class 0, and the additional experiment results are shown in Appendix D.

Table 1: The experiment results of different unlearning methods on CIFAR100 and STL10 datasets with pre-trained DDPM, and class 0 is selected to be unlearned. The best results are bolded and the second best results are underlined.

| Method | CIFAR-100 | | | | | STL10 | | | | |
|---|---|---|---|---|---|---|---|---|---|---|
| | $CE_f$ (↑) | $CE_r$ (↓) | RA (↑) | RT (↓) | FID (↓) | $CE_f$ (↑) | $CE_r$ (↓) | RA (↑) | RT (↓) | FID (↓) |
| FT (Warnecke et al., 2021) | 1.247 | 1.946 | 0.245 | 2000 | 298.6 | 1.631 | 1.628 | 0.104 | 2000 | 187.4 |
| GA (Thudi et al., 2022) | 1.222 | 1.499 | 0.239 | 2000 | 40.44 | 1.749 | 1.722 | 0.120 | 2000 | 332.7 |
| SA (Heng & Soh, 2024) | 1.306 | 1.463 | 0.401 | 20000 | 40.28 | 1.822 | 0.089 | 0.968 | 30000 | 48.87 |
| SalUn (Fan et al., 2024) | 1.218 | 1.482 | 0.381 | 2000 | 59.28 | 0.596 | 0.092 | **0.990** | 4000 | 75.91 |
| EBU (Ours) | **1.431** | **1.398** | **0.419** | **200** | **37.88** | **1.917** | **0.086** | 0.955 | **200** | **48.35** |

It can be seen from Table 1 that our EBU demonstrates a significant reduction in RT while maintaining performance on par with other baseline methods. Notably, our approach achieves superior results across most evaluation metrics, emphasizing both its efficiency and robustness. Moreover, EBU excels in CIFAR100 datasets with a higher number of classes, where the intricate correlations between the data to forget and the data to remember are more effectively managed. This highlights our method's ability to handle the optimization conflicts between the remembering and forgetting, improving the efficiency and ensuring reliable and scalable unlearning performance.

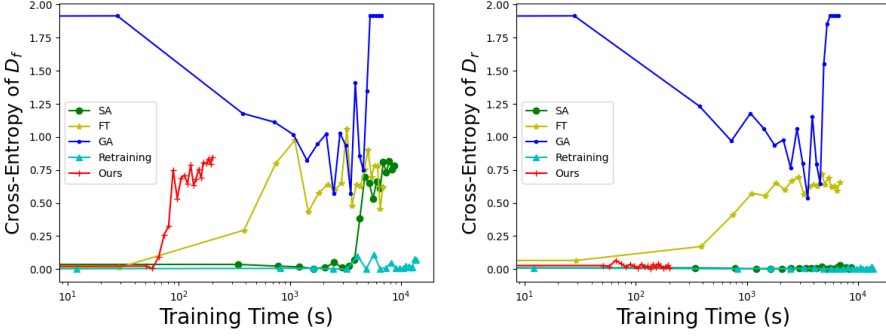

(a) CE of $D_f$ varies with unlearning time.  (b) CE of $D_r$ varies with unlearning time.

Figure 2: The changes of cross-entropy (CE) of $D_f$ and $D_r$ with different unlearning methods when unlearning time increases.

To further demonstrate the efficiency of our proposed method, we plotted the changes in classification cross-entropy of CIFAR10 when unlearning class 0 during the unlearning process in Figure 5 (more details are presented in Appendix D.2). It's evident that our method can forget the designated categories within a short time (within $10^2$ seconds) without affecting the remaining data. Moreover,

to further validate the presence of optimization conflicts on shared parameters, we report the average number of parameters related to both forgetting and remembering during the fine-tuning process in Appendix D.1.

**Main results on SD** We perform the class-wise forgetting on pre-trained standard SD model, ten classes of Imagenette are chosen to evaluate the performance of unlearning from SD. We present the experiment results of forgetting class 'cassette player' in Table 2 , and the detailed experiment settings can be found in Appendix B. We further present the generated samples of different unlearning methods with prompt "An image of cassette player" in Figure 9 (refer to Appendix D.4).

It can be seen from Figure 9 that, compared with SA, our proposed method drastically reduces the time required for forgetting, with only 1000 steps required to achieve good results whereas SA requires 50000 steps for complete forgetting, resulting in $50\times$ efficiency improvement, also compared with ESD and SalUn, our method has better unlearning perfor-

Table 2: Experiment results of different unlearning methods on pre-trained SD, class 'cassette player' is selected to be unlearned. The best results are bolded and the second best results are underlined.

| Method | Imagenette | | | | |
| | $CE_f$ (↑) | $CE_r$ (↓) | RA (↑) | RT (↓) | FID (↓) |
|---|---|---|---|---|---|
| ESD (Gandikota et al., 2023). | $1.089_{\pm.120}$ | $0.159_{\pm.022}$ | $0.936_{\pm.010}$ | **1000** | $\underline{201.9}_{\pm2.33}$ |
| SA (Heng & Soh, 2024) | $\mathbf{1.196}_{\pm.084}$ | $\mathbf{0.027}_{\pm.002}$ | $\underline{0.998}_{\pm.001}$ | 50000 | $211.7_{\pm4.01}$ |
| SalUn (Fan et al., 2024) | $1.139_{\pm.024}$ | $0.054_{\pm.032}$ | $0.976_{\pm.011}$ | **1000** | $\underline{201.7}_{\pm2.11}$ |
| EBU (Ours) | $\underline{1.148}_{\pm.054}$ | $\underline{0.041}_{\pm.005}$ | $\mathbf{0.999}_{\pm.001}$ | **1000** | $\mathbf{199.9}_{\pm2.01}$ |

mance with same gradient descent steps and better preserves the model's ability to generate samples of data to remember. The experiment results demonstrate the effectiveness and efficiency of our proposed methods in unlearning from deep generative models.

**Unlearning process visualization** We visualize the unlearning process of forgetting class 0 from the pre-trained DDPM in Figure 6 (refer to Appendix D.3), displaying a total of 200 steps. The samples of all baseline methods can be found in Appendix D. Our proposed method exhibits superior forgetting performance with the fewest forgetting time steps.

### 5.3 CONCEPT-WISE FORGETTING

Concept-wise forgetting involves the unlearning of specific concepts, often employed in the text-to-image models. In this study, we evaluate the concept-wise forgetting performance of various unlearning methods on the classic text-to-image model, SD. Our methodology begins by generating samples with empty prompts. Subsequently, we establish the *mandatory distribution* of samples for specific concepts as the distribution of these randomly generated samples.

**Forgetting Nudity** We assess the effectiveness of our EBU in forgetting nudity, the quantitative results are presented in Table 3. Moreover, we illustrate the performance of unlearning nudity in Figure 3 (more samples can refer to Appendix D.5). To ensure fair comparison across

Table 3: The experiment results of unlearning "nudity" on SD model, four baseline methods are used.

| Metric | **EBU** | SA | ESD | SalUn | SPM |
|---|---|---|---|---|---|
| Clip Score | **0.1900** | 0.1747 | 0.1895 | 0.1396 | 0.1874 |
| Nudity Score | **0.5988** | 0.4615 | 0.3557 | 0.5062 | 0.5528 |

experiments, we set the gradient descent steps to 1000 for all unlearning methods. Using the prompt "a person with full nudity," we generate samples with different seeds.

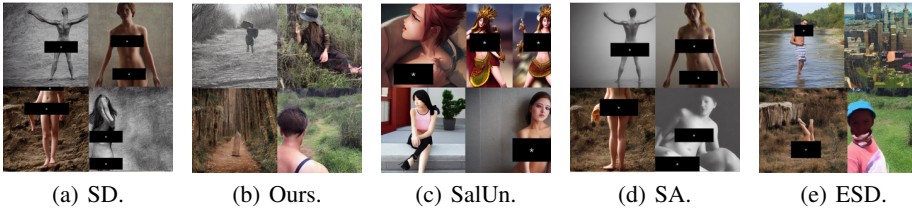

(a) SD.  (b) Ours.  (c) SalUn.  (d) SA.  (e) ESD.

Figure 3: The generated samples of nude person with prompt "a person with full nudity".

Comparative analysis with baseline methods shows that our approach consistently achieves the lowest levels of nudity, even though the number of gradient descent steps remains the same. This demonstrates the effectiveness of our method in selectively unlearning undesired content while maintaining efficiency. We also evaluate the performance of **forgetting art style** in Appendix D.5.

## 5.4 ABLATION STUDY

We have also conducted an ablation study of our proposed EBU. To evaluate the effectiveness of the Partial Align (PA) module, we remove the PA module (denoted as $\text{EBU}_{w/o\,\text{PA}}$). As for the effectiveness of Optimization Conflicts Mitigation module, we remove the $\mathcal{L}_f$, $\mathcal{L}_r^c$ and the mitigation operation respectively. The ablation study is performed on CIFAR10 dataset and the results are presented in Table 4. As shown in Table 4, the

Table 4: The ablation study of our EBU method on the CIFAR-10 dataset using DDPM.

| Ablation | $\text{CE}_f$ (↑) | $\text{CE}_r$ (↓) | RA (↑) | RT (↓) | FID (↓) |
|---|---|---|---|---|---|
| **Partial Align** | | | | | |
| $\text{EBU}_{w/o\,\text{PA}}$ | 0.8517 | 0.0353 | 0.9625 | 200 | 33.78 |
| **Optimization Conflicts Mitigation** | | | | | |
| $\text{EBU}_{w/o\,\mathcal{L}_f}$ | 0.8213 | 0.0224 | 0.9888 | 200 | 35.38 |
| $\text{EBU}_{w/o\,\mathcal{L}_r^c}$ | 0.8212 | 0.0227 | 0.9886 | 200 | 35.37 |
| $\text{EBU}_{w/o\,\text{Mitigation}}$ | 0.8033 | 0.0386 | 0.9888 | 200 | 42.42 |
| EBU | **0.8550** | **0.0110** | **0.9931** | 200 | **29.92** |

removal of the Proximal Attention (PA) significantly degrades the forgetting performance, underscoring its critical role. Furthermore, the contributions of both the forgetting loss $\mathcal{L}_f$ and the reconstruction loss $\mathcal{L}_r$ are evident in improving the overall forgetting. The proposed optimization conflict mitigation mechanism effectively reduces conflicts between objectives, leading to enhanced forgetting performance. These results demonstrate that both PA and conflict mitigation are essential components for optimizing the forgetting process.

## 5.5 EFFECT OF $\alpha$ AND $\delta$

We assess the impact of $\alpha$ and $\delta$ on the forgetting process by varying their values within the range $0.2, 0.4, 0.6, 0.8$. Our experiments are conducted on CIFAR10, and results are depicted in Figure 4. **The effect of $\alpha$**: We observe that increasing $\alpha$ affects the generated samples of both data to remember

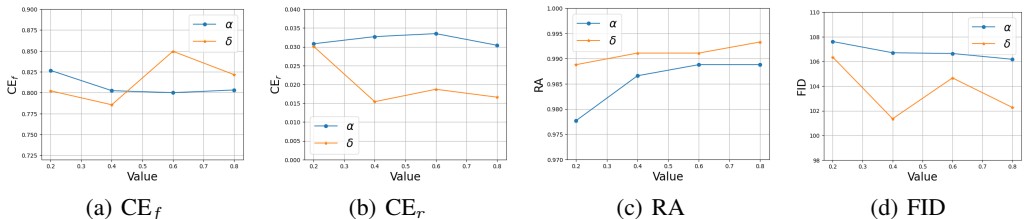

| (a) $\text{CE}_f$ | (b) $\text{CE}_r$ | (c) RA | (d) FID |

Figure 4: We investigate the impact of $\alpha$ and $\delta$ on the forgetting performance through experiments conducted on the CIFAR10 dataset. We vary the values of $\alpha$ and $\delta$ within the range $0.2, 0.4, 0.6, 0.8$.

and data to forget simultaneously. Specifically, within a small range of $\alpha$, the quality of generated samples for data to remember improves. Conversely, within a larger range, the performance of forgetting is enhanced. **The effect of $\delta$**: Furthermore, the value of $\delta$ determines the proportion of parameters attributed to forgetting and remembering. As $\delta$ increases, the quality of generated samples for data to remember improves. However, when $\delta$ reaches a large value, the forgetting process is impacted adversely due to the involvement of more irrelevant parameters.

## 6 CONCLUSION AND LIMITATION

In conclusion, we addressed the inefficiencies existed in Bayesian-based unlearning methods for deep generative models. We proposed an **E**fficient **B**ayesian-based **U**nlearning method (**EBU**), which significantly enhances the unlearning process. By pinpointing relevant parameters and balancing gradient descent directions of data to forget and data to remember, EBU preserves essential parameters and manages conflicts effectively, resulting in a more efficient unlearning process. Extensive experiments across various generative models and unlearning tasks demonstrate the superior performance of EBU, validating its effectiveness and efficiency in unlearning tasks.

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

## A    RELATED WORK OF MACHINE UNLEARNING

Machine unlearning dedicates to enabling the removal of specific data from trained machine learning models (Bourtoule et al., 2021; Nguyen et al., 2022), encompasses methods ranging from data reorganization to direct model manipulation. While some approaches focus on preventing model learning by reorganizing data or constructing fake data (Felps et al., 2020; Tarun et al., 2023; Zhang et al., 2022; Cao & Yang, 2015), they often introduce extra information that impacts learning. Some researchers tried to unlearn sample through manipulating the model directly (Baumhauer et al., 2022; Golatkar et al., 2020; Sekhari et al., 2021; Wang et al., 2024; 2023b), *i.e.*, modifying weights to remove sample information or adjusting model updates based on data statistics during training (Golatkar et al., 2020; Sekhari et al., 2021). Notably, research primarily focuses on unlearning for classification models, with relatively less attention to generative models, which is still in its early stages.

## B    EXPERIMENT SETTING

### B.1    THE DESCRIPTION OF BASELINE METHODS

**General unlearning methods:** We choose Retraining, FT (Warnecke et al., 2021) and GA (Thudi et al., 2022) as basic unlearning baseline methods. Retraining realizes unlearning by updating on the training data with removal of data. FT builds on close-form updates of model parameters to unlearn the features and labels. GA limits the overall change in weights during SGD to facilitate the approximate unlearning.

**Unlearning for generative models:** We use two most recent unlearning methods for generative models SA (Heng & Soh, 2024) and ESD (Gandikota et al., 2023). SA derives from continual learning to selectively forget concepts in pretrained deep generative models. ESD erases a visual concept from a pre-trained diffusion model, given only the name of the style and using negative guidance as a teacher model.

### B.2    IMPLEMENT DETAILS

**Class-wise forgetting:** Class-wise forgetting targets the removal of generations belonging to specified classes from deep generative models. For the DDPM, we fine-tune it for 200 gradient update steps using a batch size of 32 and a learning rate of $1e$-5. We conduct comparisons on two datasets, CIFAR-10 and STL10, employing various methods. Regarding the SD model, we fine-tune it for 1000 gradient update steps with a learning rate of $1e$-5. We focus on unlearning classes from Imagenette, which comprises ten easily identifiable classes from ImageNet. The detailed experiment results can be found in Section 5.2.

**Concept-wise forgetting:** The concept-wise unlearning of deep generative models aims to eliminate generations containing specific concepts. We fine-tune the SD model using 1000 gradient descent steps, with a batch size set to 1 and a learning rate of $1e$-5. We consider two types of concepts: art style and nudity. Detailed experiments and results can be found in Section 5.3.

### B.3    THE DETAILS OF FORGETTING FROM DDPM

**Baseline implement.** For the baseline methods, we implement them as recommend. Note that the baseline methods FT, GA are proposed for classification model, but they can be adapted to generative model easily. For FT, it only needs data to remember during the unlearning process, thus we change the original loss function of FT as $\mathcal{L}_r$. For GA, it only needs data to forget, thus we set the loss function of GA as $-\mathcal{L}_f$. And for Retraining, we just fine-tune the pre-trained model with $D_r$ directly. For SA, we implement it as recommend.

**Experiment settings** In our experimental setup, we utilize a simple yet effective Residual model architecture with 3 input and output channels, employing a channel size of 128 and 2 residual blocks. Attention resolutions are set at 16, with dropout probability at 0.1. For diffusion, we implement a linear beta schedule spanning from 0.0001 to 0.02 over 1000 diffusion timesteps. During training, we

employ a batch size of 32 for 20,0 iterations, with logging every 50 iterations and visualization of 100 samples. Optimization is conducted using the Adam optimizer with a learning rate of 0.0001 and a weight decay of 0.000, while gradients are clipped at a threshold of 1.0. These settings ensure robust experimentation and reliable evaluation of our proposed methods.

### B.4 THE EXPERIMENT SETTINGS OF FORGETTING FROM SD.

**Baseline implement** We use two baseline methods ESD and SD. For SD, it needs to generate the data to forget and data to remember, we use the random samples generated by SD as the data to forget and we then use a empty prompt to generate the data to remember. For ESD, we implement it as recommended.

**Experiment settings** In our experiment setup, we utilize the Latent Diffusion model with specific configurations tailored for unlearning tasks. The diffusion process spans 1000 timesteps, with linear beta scheduling from 0.00085 to 0.012. We employ a UNet model architecture with attention resolutions at [4, 2, 1] and two residual blocks. Training involves a base learning rate of 1.0e-05, and we utilize a LambdaLinear scheduler with a warm-up period of 1 step. The model consists of a first stage autoencoder with embedded dimensions of 4 and a conditional stage encoder. And all the unlearning methods are trained with 'xattn' part parameters while keeping other parts frozen.

## C THEORY PROOF

### C.1 PAC ASSUMPTION

To give the bound between the solution of multi-task learning problem, the following assumptions need to be introduced in advance:

**Assumption C.1.** If function $G_{\boldsymbol{\theta}}$ is probably approximately correct, for all $\boldsymbol{\theta} \in \Theta$ with probability $1 - \delta$ over independent draws $(X_n, Y_n) \sim p$:

$$|\mathbb{E}[\mathcal{L}(X, \boldsymbol{\theta})] - \frac{1}{N} \sum_{i=1}^{N} \mathcal{L}(X_n, \boldsymbol{\theta})| \leq \epsilon(N, \delta) \tag{14}$$

**Assumption C.2.** Loss function $\mathcal{L}(., \boldsymbol{\theta})$ is M-Lipschitz continuous.

The Assumption C.1 is a generalization of the law of large numbers for the case in which samples are iid, where the error is of under $1/N$. And under the assumption, we could obtain the proposition:

**Proposition C.3.** *The bounds between the optimization objective $\mathcal{O}_1$ and $\mathcal{O}_2$:*

$$|\mathcal{O}_1 - \mathcal{O}_2| \leq \mathbb{E}_{p(D_f)}[G_{\boldsymbol{\theta}_f}(X_f|Y_f) - G_{\boldsymbol{\theta}_e}(X_f|Y_f)] + \mathbb{E}_{p(D_r)}[G_{\boldsymbol{\theta}_r}(X_r|Y_r) - G_{\boldsymbol{\theta}_e}(X_r|Y_r)]$$
$$+ \epsilon(N_f + N_r, \zeta) \tag{15}$$

*where $\boldsymbol{\theta}_e$ denotes the optimal solution of Equation (2) and $G_{\boldsymbol{\theta}_{(.)}}$ denotes the deep generative model with parameters $\boldsymbol{\theta}_{(.)}$.*

The Equation (15) in the proposition is a direct application of Assumption C.1 over the average probability distribution and taking the Lipschitz Assumption C.2 over the solutions.

Note that if the function is Lipschitz in the parameterization, there is a connection between the functional, and parametric constraints:

**Proposition C.4.** *The two generative models can be seen as two parametric functions, thus it has:*

$$\mathbb{E}_{p(D_f)}[G_{\boldsymbol{\theta}_f}(X_f|Y_f) - G_{\boldsymbol{\theta}_e}(X_f|Y_f)] \leq L|\boldsymbol{\theta}_f - \boldsymbol{\theta}_e|,$$
$$\mathbb{E}_{p(D_r)}[G_{\boldsymbol{\theta}_r}(X_r|Y_r) - G_{\boldsymbol{\theta}_e}(X_r|Y_r)] \leq L|\boldsymbol{\theta}_r - \boldsymbol{\theta}_e| \tag{16}$$

Through enforcing the constraint over the parameters, we could remove the expectation and the dependency over the distribution $p(\cdot)$.

## C.2 THE PAC-BAYESIAN THEORY

PAC-Bayesian theory (McAllester, 1998) seeks to quantify the trade-off between empirical risk minimization and model complexity, offering insights into the generalization ability of a learning algorithm. Given a hypothesis class $\mathcal{H}$, the training set $D = \{(X_1, Y_1), \ldots, (X_n, Y_n)\}$ iid sampled from an distribution $\hat{p}$ over an instance space $\mathcal{S}$, the real-valued loss function $\mathcal{L} : \mathcal{H} \times \mathcal{S} \longrightarrow [0, \infty)$, the PAC-Bayes provides the generalization bounds for any posterior $q \in \mathcal{M}_+^1(\mathcal{H})$, and $\mathcal{M}_+^1(\mathcal{H})$ is the set of probability measures on a space $\mathcal{H}$. The generalization bounds are dependent on the empirical performance of $q$ and its closeness to a chosen prior distribution $p$, the empirical risks of a posterior distribution $q$ are defined as:

$$\mathcal{R}_s(q) = \mathbb{E}_{h \sim q(h)}\left[\frac{1}{n}\sum_{i=1}^{n} \mathcal{L}(h, (X_i, Y_i))\right] \tag{17}$$

and the true risk of $q$ is defined as:

$$\mathcal{R}(q) = \mathbb{E}_{h \sim q(h)}[\mathbb{E}_{(X,Y) \sim \hat{p}}\mathcal{L}(h, (X, Y))] \tag{18}$$

Also before introducing the PAC-Bayesian bound, the posterior $q$ and the loss function $\mathcal{L}$ need to satisfy the Assumption C.5.

**Assumption C.5.** *If there exists a constant $K > 0$ and a family $\mathcal{E}$ of functions $\mathcal{H} \to \mathbb{R}$, for any $(X_1, Y_1), (X_2, Y_2) \in \mathcal{S}$:*

$$d_{\mathcal{E}}(q(h|(X_1, Y_1)), q(h|(X_2, Y_2))) \geq Kd((X_1, Y_1), (X_2, Y_2)) \tag{19}$$

and the loss function $\mathcal{L}(., (X, Y)) : \mathcal{H} \to \mathbb{R}$ is in $\mathcal{E}$.

Then for the bounded loss functions, the PAC-Bayesian bound (Catoni, 2003) are defined in Theorem C.6

**Theorem C.6.** *For a probability measure $\hat{p}$ on $\mathcal{S}$, a loss function $\mathcal{L} : \mathcal{H} \times \mathcal{S} \longrightarrow [0, 1]$, with probability at least 1-$\delta$ over the $n$ random samples $\hat{S}$ draw from $\hat{p}$, the following equation holds for any posterior distribution $q \in \mathcal{M}_+^1(H)$:*

$$\mathcal{R}(q) \leq \mathcal{R}_s(q) + \frac{\lambda}{8n} + \frac{D_{KL}(q||p) + log\frac{1}{\delta}}{\lambda} \tag{20}$$

*the real number $\delta \in (0, 1)$ and $\lambda > 0$.*

The Theorem C.6 predicts the behavior of $q(h|(X, Y))$ for any $(X, Y) \sim \hat{p}$ when the posterior $q$ was learned using the training samples $\mathcal{D}$.

# D EXPERIMENT RESULTS

## D.1 THE COUNT OF OVERLAPPED PARAMETERS

We have counted the average number of selected forgetting and remembering parameters of the SD model, as well as the overlapping parameters during the unlearning process. In our experiments,

Table 5: The average count of parameters related to forgetting and remembering and the overlapped parameters of SD model during the unlearning process.

| Forgetting concept | Total | Forgetting | remembering | overlapped |
|---|---|---|---|---|
| Multiple Artists | 3.200G | 1.605G | 1.988G | 1.095G |
| Single Artist | 3.200G | 1.701G | 1.596G | 0.867G |
| Multiple violence concepts | 3.200G | 1.616G | 2.010G | 1.100G |
| Single violence concepts | 3.200G | 1.675G | 1.566G | 0.861G |

we observed that there is indeed a non-negligible overlap between the parameters $\theta_r$ (retain) and $\theta_f$ (forget), and we present the count of overlapped parameters in Table 5 . This overlap was identified through the gradient analysis during the training process, where conflicting updates to shared parameters were detected.

## D.2 EFFICIENCY ANALYSIS

To further demonstrate the efficiency of our proposed method, we plotted the changes in classification cross-entropy of CIFAR10 when unlearning class 0 during the unlearning process in Figure 5. It's evident from the plot that our method can forget the designated categories within a short time (within $10^2$ seconds) without affecting the remaining data. In contrast, other methods consistently require more than $10^3$ seconds for the same task. This evidence highlights that our method achieves a significant improvement in unlearning time and is more efficient than other baseline methods.

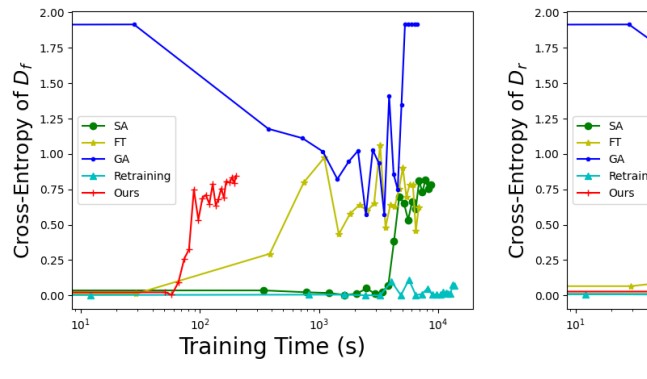

(a) CE of $D_f$ varies with unlearning time.    (b) CE of $D_r$ varies with unlearning time.

Figure 5: The changes of cross-entropy (CE) of $D_f$ and $D_r$ with different unlearning methods when unlearning time increases, experiments are performed on CIFAR10 dataset, class 0 is unlearned.

## D.3 UNLEARNING PROCESS

We present the visualization of unlearning process of the unlearning methods with 200 gradient descent steps on CIFAR10 dataset in figure 6.

## D.4 CLASS-WISE FORGETTING

**Forgetting from pre-trained DDPM.** The full experiment results on CIFAR10 dataset are presented in Table 6:

Table 6: The experiment results of different unlearning methods on CIFAR10 and STL10 datasets with pre-trained DDPM, and class 0 is selected to be unlearned. The best results are bolded and the second best results are underlined.

| Method | CIFAR-10 | | | | | STL10 | | | | |
|---|---|---|---|---|---|---|---|---|---|---|
| | CE$_f$ (↑) | CE$_r$ (↓) | RA (↑) | RT (↓) | FID (↓) | CE$_f$ (↑) | CE$_r$ (↓) | RA (↑) | RT (↓) | FID (↓) |
| FT (Warnecke et al., 2021) | $0.579_{\pm.006}$ | $0.580_{\pm.001}$ | $0.114_{\pm.001}$ | $\underline{2000}$ | $75.51_{\pm21.1}$ | $1.631_{\pm.162}$ | $1.628_{\pm.191}$ | $0.104_{\pm.005}$ | $\underline{2000}$ | $187.4_{\pm21.2}$ |
| GA (Thudi et al., 2022) | $0.627_{\pm.005}$ | $0.609_{\pm.005}$ | $0.596_{\pm.008}$ | $\underline{2000}$ | $253.6_{\pm15.0}$ | $1.749_{\pm.009}$ | $1.722_{\pm.007}$ | $0.120_{\pm.024}$ | $\underline{2000}$ | $332.7_{\pm10.1}$ |
| SA (Heng & Soh, 2024) | $\underline{0.807}_{\pm.006}$ | $\underline{0.009}_{\pm.001}$ | $\mathbf{0.997}_{\pm.001}$ | $20000$ | $\mathbf{19.11}_{\pm2.41}$ | $\underline{1.822}_{\pm.284}$ | $0.089_{\pm.005}$ | $\underline{0.968}_{\pm.013}$ | $30000$ | $48.87_{\pm2.81}$ |
| SalUn (Fan et al., 2024) | $0.598_{\pm.012}$ | $0.061_{\pm.004}$ | $0.996_{\pm.003}$ | $4000$ | $29.91_{\pm2.12}$ | $0.596_{\pm.018}$ | $0.092_{\pm.006}$ | $\mathbf{0.990}_{\pm.015}$ | $4000$ | $75.91_{\pm2.83}$ |
| EBU (Ours) | $\mathbf{0.855}_{\pm.051}$ | $0.011_{\pm.001}$ | $0.993_{\pm.001}$ | $\mathbf{200}$ | $\underline{29.92}_{\pm4.21}$ | $\mathbf{1.917}_{\pm.021}$ | $\underline{0.086}_{\pm.015}$ | $0.955_{\pm.021}$ | $\mathbf{200}$ | $\underline{48.35}_{\pm4.32}$ |

The visualizations of different unlearning methods unlearn from pre-trained DDPM on CIFAR10 and STL10 are presented on figure 7 and figure 8.

**Forgetting from pre-trained SD.** We present the generated samples of different unlearning methods with prompt "An image of cassette player" in Figure 9.

## D.5 CONCEPT-WISE FORGETTING

**Forgetting of Nudity.** We illustrate the performance of unlearning nudity of four baseline methods in Figure 10.

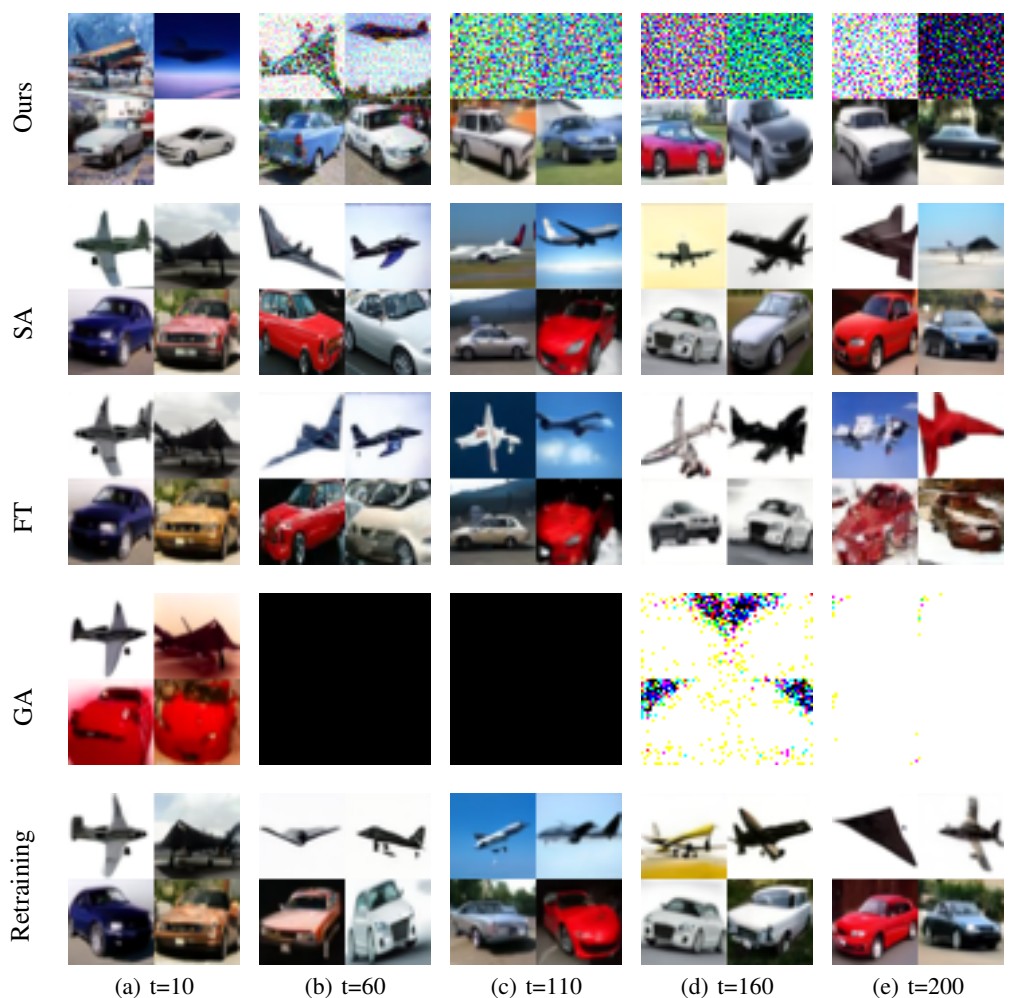

(a) t=10  (b) t=60  (c) t=110  (d) t=160  (e) t=200

Figure 6: Samples generated by our method, SA, and FT for the classes 0 ("airplane") and class 1 ("car") on the CIFAR10 dataset. The class to forget is "airplane", and the class to remember is "car". The unlearning step 't' varies from 10 to 200.

**Forgetting Art style** To assess the efficacy of various forgetting methods in unlearning art styles, we conduct experiments using the SD model. In Figure 11, we present the forgetting results for "Kelly Mckernan" and "Thomas Kinkade". Our EBU demonstrates the capability to effectively forget art styles from the generative model. The figures generated by our method exhibit different colors and objects compared to the original figures generated by the SD model. This showcases the effectiveness of EBU in forgetting art styles.

## D.6 Effect of Hyper-Parameter

The impact of $\alpha$ and $\delta$ on the forgetting process, their values are chosen in the range $0.2, 0.4, 0.6, 0.8$. Our experiments are conducted on CIFAR10, and results are depicted in Figure 12.

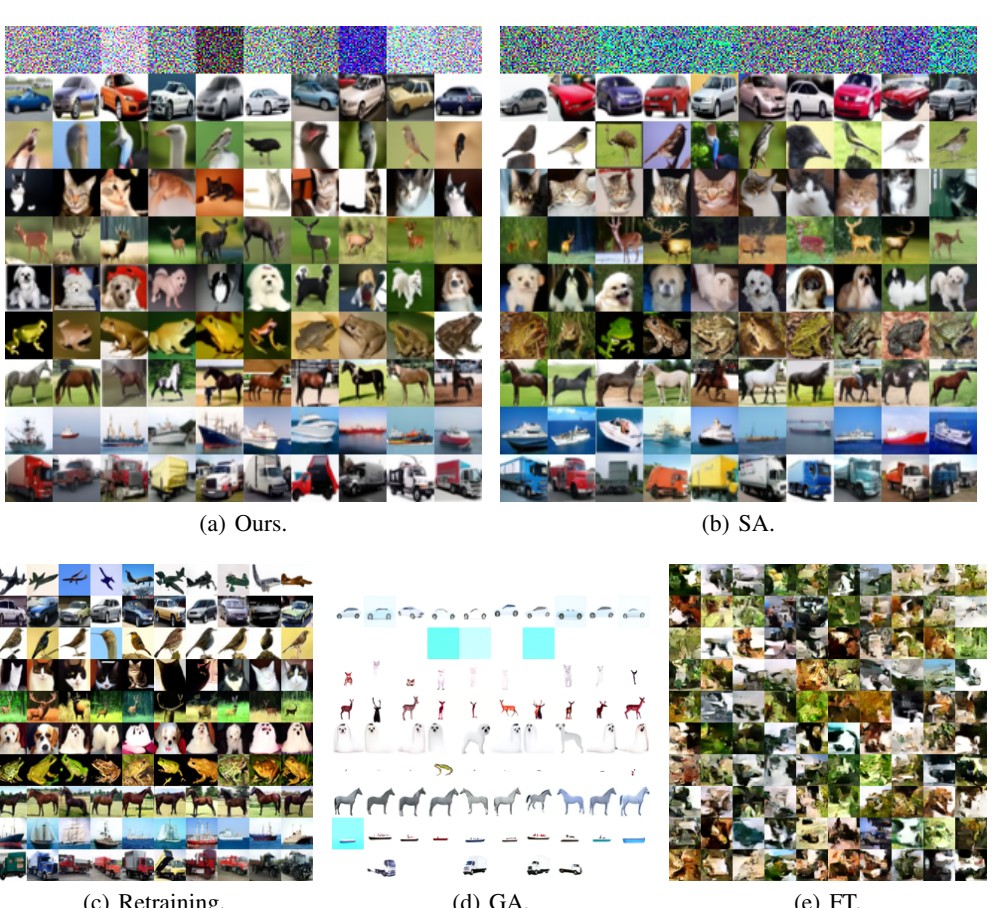

(a) Ours.

(b) SA.

(c) Retraining.

(d) GA.

(e) FT.

Figure 7: Unlearn class 0 on CIFAR10 Dataset.

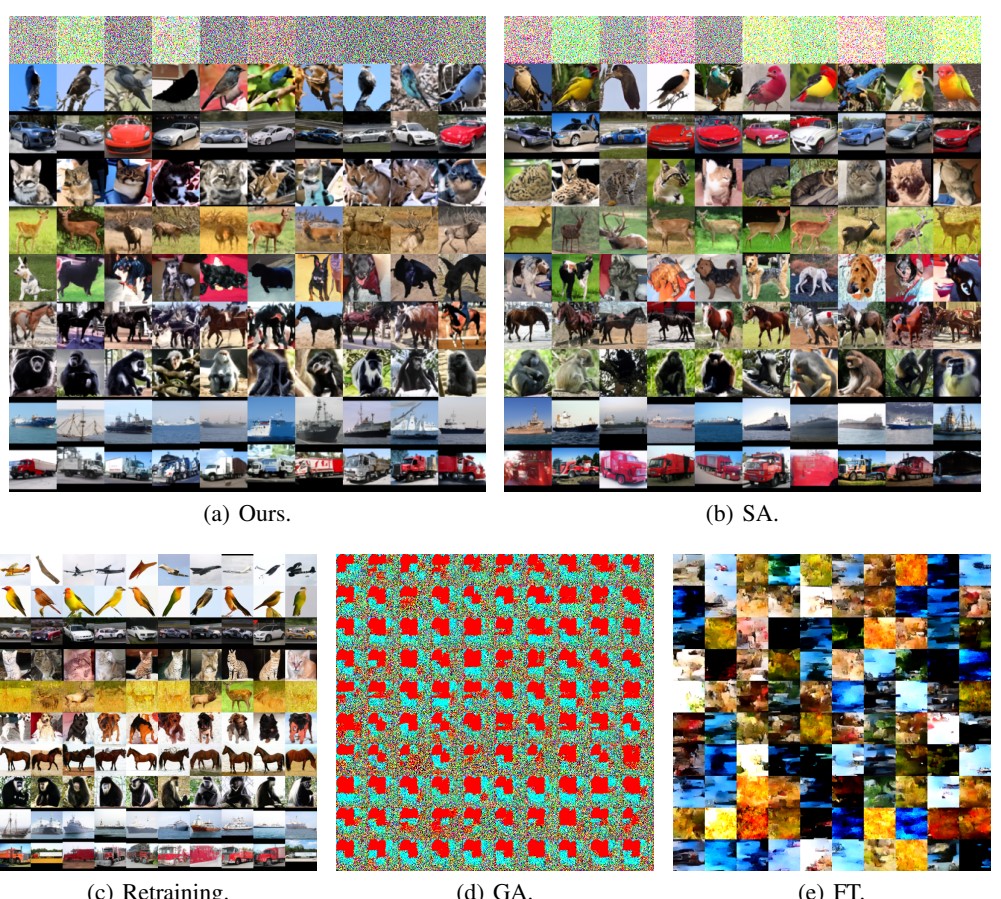

(a) Ours.      (b) SA.

(c) Retraining.      (d) GA.      (e) FT.

Figure 8: Unlearn class 0 on STL10 Dataset.

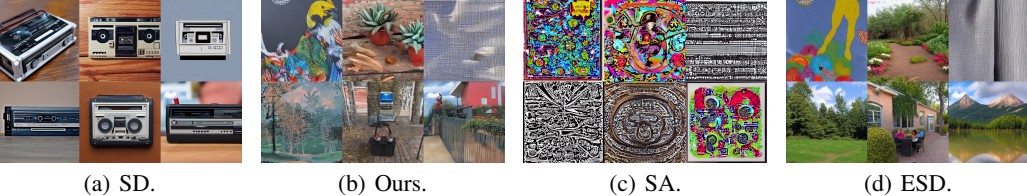

(a) SD.      (b) Ours.      (c) SA.      (d) ESD.

Figure 9: Generated samples of object "cassette player" with prompt "An image of cassette player".

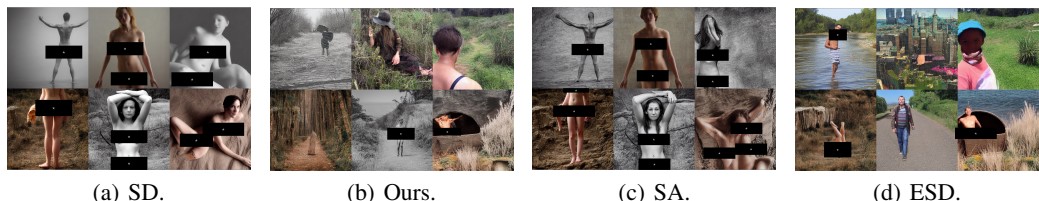

(a) SD.      (b) Ours.      (c) SA.      (d) ESD.

Figure 10: The generated samples of nude person with prompt "a person with full nudity".

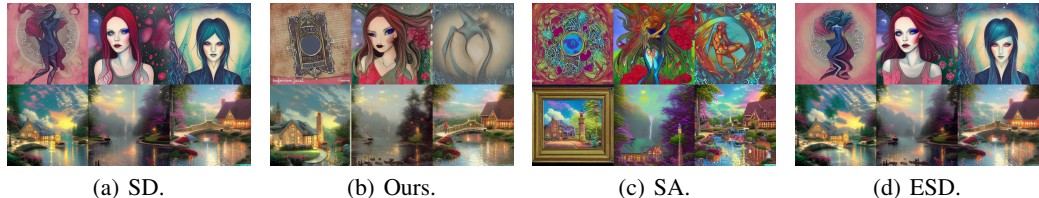

(a) SD.  (b) Ours.  (c) SA.  (d) ESD.

Figure 11: The generated samples of art style 'Kelly Mckernan' (top line) and 'Thomas Kinkade' (bottom line) with prompt "An image of Kelly Mckernan" and "An image of Thomas Kinkade".

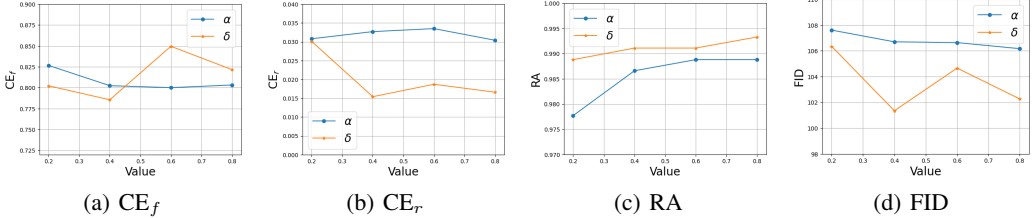

(a) $CE_f$  (b) $CE_r$  (c) RA  (d) FID

Figure 12: We investigate the impact of $\alpha$ and $\delta$ on the forgetting performance through experiments conducted on the CIFAR10 dataset. We vary the values of $\alpha$ and $\delta$ within the range $0.2, 0.4, 0.6, 0.8$.

