# OpenReview forum: "Efficient Machine Unlearning for Deep Generative Models by Mitigating Optimization Conflicts"
_ICLR.cc/2025/Conference — ICLR 2025 Conference Withdrawn Submission_

### Official Review · Reviewer_Z19w · 2024-10-27

**Soundness:** 2
**Presentation:** 1
**Contribution:** 2
**Rating:** 3
**Confidence:** 4

**Summary:**

This work proposes an unlearning method called EBU. This method dynamically selects parameters specifically related to forgetting and remembering during the fine-tuning process, which makes the unlearning process more efficient. Besides, EBU balances gradient updates on shared parameters associated with both types of data by, considering the correlation between data to forget and data to remember. Extensive experiments have been conducted to show the effectiveness of EBU.

**Strengths:**

1. The motivation and method description of EBU are clear.
2. The experiments are solid in validating part of the arguments proposed in the method. For example, Fig. 2 is good evidence of preserving $D_r$.
3. The qualitative results in concept erasing look good.

**Weaknesses:**

1. The writing is confusing. Here is one example, in lines 138 - 140, are $\theta_f$ and $\theta_r$ different models or subsets of one single model? Based on later analysis they seem to be subsets of one model. Besides, the authors wrote ''Fast forgetting of $D_f$ can be achieved by keeping $θ_r$ consistent with its original values and leaving $θ_f$ unchanged''. Where does the unlearning happen if keeping both $θ_r$ and $θ_f$ either consistent with their original values or unchanged?
2. $\mathcal{L}_f$ and $\mathcal{L}_r$ are not defined until one page later than mentioned in Sec. 4.1, making the method section hard to follow.
3. In Eq. 2, $\theta$ is the trainable parameter, while in Eq. 7, $\theta$ becomes the original model parameters.
4. The experiment section is unpolished. For example, in Table 1's caption: ''The best results are bolded and the second best results are underlined.'', I failed to find any underlines in Table 1.
5. The author define ''UT'' as the ''unlearning time'' but never report it. Thus, I fail to tell whether the proposed method is more efficient or not.

**Questions:**

1. As defined in problem formulation, $D_f$ is not part of the training data $D$, then how to define ''forget'' in this setting? The model has never ''remembered'' $D_f$.
2. Instead of doing gradient modulation, why not just leave the overlapped parameters unchanged or update with $\nabla \mathcal{L}_f$ more mildly? What could be the potential problem compared with balancing the effects?
3. In the experiment section, why RT (relearn time) is the lower the better? A good unlearning method should make the model robust to relearning.
4. Will the performance drop on the classes/concepts other than $D_r$?

---

### Official Review · Reviewer_Z8gC · 2024-10-28

**Soundness:** 2
**Presentation:** 2
**Contribution:** 3
**Rating:** 5
**Confidence:** 2

**Summary:**

This paper proposes a new efficient method for Bayesian-based unlearning of generative models. It identifies the parameters corresponding to the data for forgetting and selectively retains the other parameters.  It also balances gradient updates by considering the overlap between the parameters for forgetting and the others. Specifically, the parameters corresponding to the data for forgetting are optimized to minimize the KL divergence between the conditional probability given the label for forgetting and the normal distribution, while the other parameters are optimized to minimize the KL divergence with the initial generative model. The selection of these parameters is done by using top-k with respect to the gradient of the loss. Since the parameters for forgetting and the parameters for remembering can be overlapped, the gradients of each objective function are balanced by using an interior point.

**Strengths:**

1. Unlearning is an important research field, and the proposed method seems to be the fundamental technique that can be used in moderately broad cases.
1. The proposal seems reasonable. However, it seems a little straightforward and there are not many big surprises.
1. The experimental results demonstrate that the proposed method outperforms baselines in terms of the metric of forgetting and the metric of generating models for data to be remembered.
 In addition, efficiency is measured by runtime, and it is impressively fast compared to baselines. Multiple datasets and models are used for evaluation, and sensitivity to hyperparameters for the proposed method is also evaluated.

**Weaknesses:**

1. The method is a bit straightforward and simple. There is not much surprise in the gradient-based parameter selection and two objective functions for forgetting and remembering.
Since the theoretical contributions are also weak, as shown below, the paper would be stronger if there was more strong theoretical results or more experimental analysis including new insights.

1. The mathematical notation is unclear, and the correctness and significance of the theoretical results are unclear.
In lines 138-141, it appears that $\theta_r$ and $\theta_f$ are part of the model parameter $\theta$.
However, in equation (3), the proposed method discusses the model $G_{\theta_r}$ consisting only of $\theta_r$ or $theta_f$, which does not match the previous explanation. I
If $_r$ and $_f$ do not represent part of the parameters but represent a state of the parameters, such as the optimal state for some objective, then their definition is necessary.
I could not judge the correctness and significance of Proposition 4.1 due to the above unclearness and the little explanation of proofs in the appendix.
The contribution of the derivation of Proposition 4.2 seems a bit trivial since it can be obtained immediately from the assumption of Lipschitz continuity.


1. The relationship between $\theta_r$ and $theta_f$ is also unclear in Figure 1.
In this figure, $\theta_r$ and $theta_f$ appear subspaces in the possible parameter space.
However, originally, $theta_r$ and $theta$ and defined as a vector, not as a set of vectors.
If $\theta$ represents a set of vectors, what is $G_theta$?
If this paper justifies the proposed method using mathematical formulas, I think that it requires a clear definition and a discussion.

**Questions:**

1. I would like to see a clear definition of $theta_r$, $theta_f$, and $\theta$. How should I look at the relationship between $theta$ and $theta_r$ in Figure 1?
1. How should readers understand the relationship between the theoretical results and the proposed method?

---

### Official Review · Reviewer_ZrSw · 2024-10-30

**Soundness:** 3
**Presentation:** 2
**Contribution:** 3
**Rating:** 3
**Confidence:** 4

**Summary:**

The paper proposes a new method for machine unlearning of deep generative models. Unlike existing unlearning methods that have low efficiency, the proposed method, called EBU, improves the unlearning efficiency by identifying the weights pertaining to the data to forget and the data to remember. Experiments are conducted on the pre-trained DDPM and stable diffusion to verify the effectiveness of EBU.

**Strengths:**

The presented method for machine unlearning is new to my knowledge.

The idea of the presented method is convincing despite with some ambiguous details.

**Weaknesses:**

Some statements should be revised to improve clarity. For example, in the Abstract, the statement "EBU only preserves the parameters related to data to remember" is confusing, because EBU also trains all model parameters. In Line 102, it says "We identify these shared parameters by analyzing corresponding weight saliency maps during the unlearning process"; but the gradient information is used instead.

The notations should be carefully revised to make it easier to understand. For example, in Line 142, both $\theta_f$ and $\theta$ in $\theta_f \cap \theta=\theta_f$ are weights, not sets.

The parameters selection procedure is the foundation of the proposed method. However, empirical experiments justifying its effectiveness are lacking.

**Questions:**

In Lines 216-218, "To forget ..., we need to make the posterior distribution ... far from the real distribution ... as much as possible," why?

In Line 289, it's questionable that "During the unlearning process, the parameters that are closely related to the forgetting and remembering tasks will exhibit larger gradients compared to the irrelevant parameters. This observation ..." Specifically, the gradient of the parameters related to the remembering tasks is expected to be small, because the remembering tasks are similar to the original training tasks. This observation should be empirically demonstrated and extensively verified.

In Eq. (10), it's the absolute value of the gradient is used, isn't it? Is the parameters selection procedure performed in each iteration? Also, how is $\sigma$ set in the experiments?

It seems that the gradient in Eq. (11) is not used in Algorithm 1?

How many backpropagations are performed in each iteration?

---

### Official Review · Reviewer_FrGm · 2024-11-03

**Soundness:** 1
**Presentation:** 2
**Contribution:** 1
**Rating:** 3
**Confidence:** 4

**Summary:**

The paper studies the problem of efficiently unlearning a pretrained deep generative model on certain undesirable data while keeping the knowledge on other benign data. To solve this problem, the paper proposes a new loss function and a few optimization techniques for more efficient unlearning. The paper studies the effect of the proposed method on unlearning an entire label and unlearning concepts like nudity and art styles, and compares to several baseline models.

**Strengths:**

The paper aims to solve a very important problem in trustworthy ML that widely exists in many generative models such as image generation. By directly modifying the weights, the proposed method makes it harder to misuse even when the weights are released or shared compared to posthoc filtering methods and negative guidance based sampling algorithms.

The proposed loss functions are very intuitive, and the proposed optimization methods also seem to accelerate unlearning significantly. The effectiveness and efficiency are validated in both controlled experiments and more practical settings where concepts are to be unlearned.

**Weaknesses:**

The writing and presentation of the paper are not clear enough and confusing in many places. It looks to me different sections are written by different authors and they are not consistent with each other. For instance, $\sigma$ in section 4 is $\delta$ in section 5. $\zeta$ in section 4 is $\delta$ in C.1. $L$ in eq 5 is $M$ in C.1.

The mathematical problem of unlearning is ill-defined in this paper. This is because $\theta_f$ and $\theta_r$ are not formally defined in section 3.2. There is also no justification for why only a subset of weights are used for unlearning some data, and I do not agree with this assumption unless there is proof of existence and uniqueness of $\theta_f$ and $\theta_r$. Consequently, the losses and gradient computations are not valid to me.

The theory in section 4.1 is just simple implementation of PAC bounds and is not presented correctly. In eq 2 there should be separate averages for the two sums, and in eq 3 the $N_f+N_r$ should be the min of these two. Prop 4.1 is missing the with high probability statement. More importantly, the theory has nothing to do with the proposed loss in eq 6,7,9.

The mandatory target distribution in eq 6 is Gaussian, but there is no justification why it is the most effective one, especially when the model has to learn the same Gaussian for all $y\in Y_f$.

As for experiments in section 5.3, they are not extensive enough. While the results on the single nudity prompt look better, there is no systematic proof of unlearning of this concept because other natural language prompts or even adversarial prompts might trigger the concept to be generated. There is also lack of quantitative study for art styles and other tasks such as the I2P dataset in SLD.

**Questions:**

Please refer to the weakness section above.

---

### Note · Authors · 2024-11-14

I have read and agree with the venue's withdrawal policy on behalf of myself and my co-authors.